# The PHD finger of *Arabidopsis* SIZ1 recognizes trimethylated histone H3K4 mediating SIZ1 function and abiotic stress response

Kenji Miura [1,2,3]*, Na Renhu[1,3] & Takuya Suzaki[1,2]

*Arabidopsis* SIZ1 encodes a SUMO E3 ligase to regulate abiotic and biotic stress responses. Among SIZ1 or mammalian PIAS orthologs, plant SIZ1 proteins contain the plant homeodomain (PHD) finger, a $C_4HC_3$ zinc finger. Here, we investigated the importance of PHD of *Arabidopsis* SIZ1. The $Pro_{SIZ1}::SIZ1(\Delta PHD):GFP$ was unable to complement growth retardation, ABA hypersensitivity, and the cold-sensitive phenotype of the *siz1* mutant, but $Pro_{SIZ1}::SIZ1:GFP$ could. Substitution of C162S in the PHD finger was unable to complement the *siz1* mutation. Tri-methylated histone H3K4 (H3K4me3) was recognized by PHD, not by PHD (C162S). *WRKY70* was up-regulated in the *siz1-2* mutant and H3K4me3 accumulated at high levels in the *WRKY70* promoter. PHD interacts with ATX, which mediates methylation of histone, probably leading to suppression of ATX's function. These results suggest that the PHD finger of SIZ1 is important for recognition of the histone code and is required for SIZ1 function and transcriptional suppression.

[1] Graduate School of Life and Environmental Sciences, University of Tsukuba, Tsukuba 305-8572, Japan. [2] Tsukuba-Plant Innovation Research Center (T-PIRC), University of Tsukuba, Tsukuba 305-8572, Japan. [3]These authors contributed equally: Kenji Miura, Na Renhu. *email: miura.kenji.ga@u.tsukuba.ac.jp

Sumoylation, the covalent attachment of small ubiquitin-like modifier (SUMO) to other proteins, is a post-translational modification that controls protein function, activity, localization, and turnover in eukaryotes[1,2]. In plants, sumoylation is involved in the response to abiotic and biotic stresses, such as cold, salt, and drought stresses, and innate immunity[3–8]. Furthermore, sumoylation regulates signaling pathways for plant hormones, including abscisic acid (ABA) and salicylic acid[7,9–12]. Similar to ubiquitin, three enzymes, E1, E2, and E3, are required for the attachment of SUMO to other proteins[2]. E1, the heterodimeric SUMO-activating enzyme, which is composed of the SAE1 and SAE2 subunits, binds to SUMO via a high-energy thioester linkage[13,14]. Activated SUMO is transferred to E2, SUMO-conjugating enzyme 1 (SCE1) via transesterification, and then conjugated to substrate proteins, assisted by E3, SUMO ligase[15,16], via an isopeptide bond between the C-terminal glycine of SUMO and specific lysine(s) within the target. SUMO is often conjugated to lysine in the conserved ΨKXE/D (Ψ, a large hydrophobic amino acid; K, lysine; X, any amino acid; E, glutamate; and D, aspartate) motifs. At the molecular level, sumoylation alters the function of the targets, including changes in their intracellular localization, activity, and interaction with other proteins[17]. Previous proteomic studies have identified over 1000 SUMO1/2 targets in *Arabidopsis*[18,19]. Most of these targets are nuclear-localized proteins and these proteins have functions related to DNA modification, chromatin assembly, transcription factors, coactivators/repressors, and abiotic and biotic stress responses[19].

In *Arabidopsis*, four SUMO E3 ligases have been identified; SAP and MIZ1 domain-containing ligase1 (SIZ1)[15], methyl methansulfonate-sensitive21 (MMS21)/high ploidy2 (HPY2)[16,20], and protein inhibitors of activated STATs-like1 (PIAL1) and PIAL2[21]. All ligases contains the SP-RING (Siz/PIAS-RING) domain, which functions for SUMO E3 ligase activity. The *siz1-2 hpy2-1* double mutation causes embryonic lethality[22], indicating that these SUMO E3 ligases play important roles in sumoylation in *Arabidopsis* but they have distinct role in development[22]. Among the >1000 SUMO targets in *Arabidopsis*, few could be assigned to MMS21, whereas numerous targets could be assigned to SIZ1[19], suggesting that MMS21 modifies a small number of proteins and that both SUMO and SIZ1 are crucial regulators of chromatin function and transcription. PIAL1 and PIAL2 function as E4-type SUMO ligases to promote SUMO chain formation and are involved in salt and osmotic stress responses[23]. Among these four SUMO E3 ligases, SIZ1 has high similarity to yeast Siz and animal PIAS orthologs[24], which contain SAP and SP-RING domains and PINIT (for Pro-Ile-Asn-Ile-Thr) and SXS (Ser-any amino acid-Ser) motifs[15,25]. However, only plant SIZ proteins contain a PHD (plant homeodomain) finger, which is a $C_4HC_3$ (four cysteines, one histidine, and three cysteines) zinc-finger-like motif. The PHD finger proteins of ING2 and bromodomain and PHD finger transcription factor (BPTF) recognize trimethylated Lys4 of histone H3 (H3K4me3)[26,27]. Thus, it is thought that PHDs read the epigenetic code[28]. Investigation of the PHD finger of rice SIZ1 bound to methylated histone H3 by NMR revealed that OsSIZ1-PHD recognized both dimethylated Arg2 and trimethylated Lys4 of histone H3[29]. The PHD finger of *Arabidopsis* SIZ1 binds to AtSCE1 and is required for sumoylation of GTE3, a bromodomain protein, together with SP-RING domain, suggesting that PHD and SP-RING contribute to SUMO E3 ligase function[30]. Although the PHD finger seems to be important for biochemical function, a point mutation in the PHD of *Arabidopsis* SIZ1 complemented several *siz1-2* phenotypes, such as plant growth retardation, thermosensitive seed germination, and hypersensitivity to ABA-induced inhibition of cotyledon

expansion[25]. Conversely, point-mutated SP-RING was unable to complement the *siz1-2* phenotype[25].

To confirm biological importance of the PHD finger in SIZ1, we transformed *ProSIZ1::SIZ1(ΔPHD):GFP* into the *siz1-2* mutant. Although *ProSIZ1::SIZ1:GFP* was able to complement the *siz1* mutation[31], *ProSIZ1::SIZ1(ΔPHD):GFP* was not, suggesting the biological importance of the PHD finger of SIZ1. In addition, *ProSIZ1::SIZ1(C162S):GFP* was not able to complement it. The biochemical function of PHD is the preferential recognition of histone H3K4me3. Substitution of C162S in the PHD finger prevented recognition of histone H3K4me3, probably preventing complementation of the *siz1-2* mutation. Histone H3K4me3 is enriched with transcriptionally active promoters[32]. In human cells, recognition of H3K4me3 by ING2-PHD stabilizes the mSin3a–HDAC1 complex to repress active genes in response to DNA damage[33]. Because H3K4me3 was highly accumulated in the promoter of *WRKY70* in the *siz1-2* mutant, SIZ1 was suggested to repress active *WYKY70* gene expression via recognition of H3K4me3 by the PHD finger. ATX proteins methylate histone H3K4[34,35]. The PHD finger also interacts with ATX proteins. It is likely that PHD suppresses the methylation function of ATX proteins. In this article, we demonstrate importance of the PHD finger of SIZ1 on recognition of histone code and transcriptional suppression.

## Results

**PHD finger is important for SIZ1 function**. Plant SUMO E3 ligases, SIZs, contain several motifs and domains, such as SAP (Scaffold attachment factor A/B/acinus/PIAS), PHD, PINIT, SP-RING, SXS, and NLS[15,25]. Among them, the PHD finger is a unique domain of plant SIZ proteins, whereas SIZ/PIAS proteins in yeast and animals contain no PHD finger. In vitro analysis revealed that the PHD and SP-RING domains of *Arabidopsis* AtSIZ1 are required for binding to the AtSCE1, the SUMO E2-conjugating enzyme, and for sumoylation[30]. Thus, the PHD finger is assumed to be important for function of AtSIZ1. However, substitution of C134 to tyrosine in the PHD of AtSIZ1 was able to complement phenotypes of the *siz1-2* mutant, such as dwarf-like, thermosensitivity of seed germination, and ABA hypersensitivity[25]. Expression of AtSIZ1(C134Y) in *siz1-2* resulted in abnormal hypocotyl elongation in response to sugar and light, whereas the *siz1-2* mutant did not exhibit such a phenotype[25].

To confirm whether the PHD finger is important for SIZ1 function, *ProSIZ1::SIZ1(ΔPHD):GFP* was expressed in the *siz1-2* mutant to complement the dwarf-like phenotype. Expression of *ProSIZ1::SIZ1(ΔPHD):GFP* was unable to complement the dwarf-like phenotype of the *siz1-2* mutant, whereas *ProSIZ1::SIZ1:GFP* was (Fig. 1a). The expression of *SIZ1(ΔPHD):GFP* was confirmed by RT-PCR (Fig. 1b). The results suggest that PHD is important for complementing the dwarf-like phenotype of the *siz1-2* mutant. Then, a substitution was introduced in *ProSIZ1::SIZ1:GFP* and transformed into the *siz1-2* mutant. Because the PHD finger is a $C_4HC_3$ zinc-finger domain and cysteines and histidine are required for binding to zinc[36], *ProSIZ1::SIZ1(C117S):GFP* or *ProSIZ1::SIZ1(C162S):GFP* was expressed in the *siz1-2* mutant. Expression of *SIZ1(C117S)*, as well as *SIZ1(C134Y)*, was able to complement the dwarf-like phenotype[25], but *SIZ1(C162S)* was not (Fig. 1a). Expression of *SIZ1* variants was confirmed by RT-PCR (Fig. 1b). These results indicate that C162 in PHD is important for complementation of the *siz1-2* mutation. The amino-acid sequence of the PHD finger in SIZ1 demonstrated that C162 is in an α-helix, whereas C117 and C134 are not (Fig. 1c). It is probable that substitution of C162 to serine severely affects SIZ1 function.

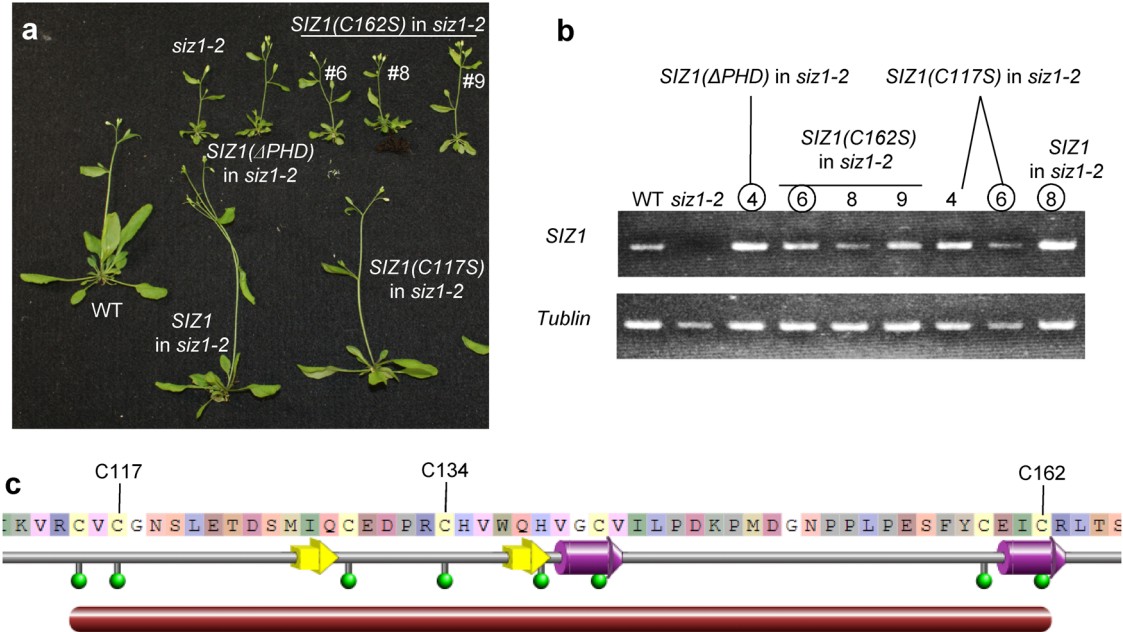

**Fig. 1 Mutation in the plant homeodomain (PHD) finger of SIZ1 was unable to complement the dwarf-like phenotype of the *siz1-2* mutant. a** *Pro_SIZ1*::SIZ1:
GFP or its variants were transformed into the *siz1-2* mutant. Six-week-old *siz1-2* mutant exhibited a dwarf-like phenotype, as previously described[39].
Introduction of *Pro_SIZ1*::SIZ1:GFP or *Pro_SIZ1*::SIZ1(C117S):GFP complemented the dwarf-like phenotype of *siz1-2*; however, introduction of *Pro_SIZ1*::SIZ1(C162S):
GFP or *Pro_SIZ1*::SIZ1(ΔPHD):GFP was unable to complement the dwarf-like phenotype of *siz1-2*. **b** Expression of *SIZ1* in each plant. *SIZ1* was not detected in the
*siz1-2* mutant. Conversely, *SIZ1* transcripts were detected in other plants. **c** Amino-acid sequence and structural information of the PHD finger in SIZ1. Green
circles indicate zinc-binding residues. Purple and yellow arrows indicate α-helices and β-sheets, respectively. The illustration was downloaded from Protein
Data Bank Japan (PDBj, https://pdbj.org/mine/structural_details/1wew).

Next, we examined whether other phenotypes of the *siz1-2*
mutant were complemented by expression of *SIZ1* variants. The
*siz1-2* mutant exhibited ABA hypersensitivity for primary root
growth, compared with wild-type seedlings (Fig. 2a and b)[37].
Expression of *SIZ1:GFP* or *SIZ1(C117S):GFP* suppressed the ABA
hypersensitivity of *siz1-2* seedlings, whereas expression of *SIZ1
(ΔPHD):GFP* or *SIZ1(C162S):GFP* did not (Fig. 2 and Supple-
mentary Fig. 1). Furthermore, expression of *SIZ1:GFP* or *SIZ1
(C117S):GFP* complemented the cold sensitivity and drought
tolerance of *siz1-2* plants, whereas expression of *SIZ1(ΔPHD):
GFP* or *SIZ1(C162S):GFP* in the *siz1-2* mutant still exhibited cold
sensitivity (Fig. 3 and Supplementary Fig. 2) and drought
tolerance (Fig. 4 and Supplementary Fig. 3), indicating that the
PHD finger is required for SIZ1 function in response to ABA,
cold stress, and drought stress.

**PHD recognizes trimethylated histone H3K4**. To elucidate the
function of the SIZ1 PHD finger, its structure was compared with
that in human and rice (Fig. 5). The structures of SIZ1 PHD
(1wew), human BPTF PHD (2fuu), and rice SIZ1 PHD (2rsd)
have been registered in the PDB database. The PHD finger of
human BPTF recognizes histone H3K4me3[27]. Furthermore, the
PHD finger of rice SIZ1 interacts with histone H3K4me3[29].
Because the structure of the PHD finger of SIZ1 is similar to that
of BPTF and that of rice SIZ1 (Fig. 5), it is assumed that the PHD
finger of SIZ1 recognizes histone H3K4me3.

The GST-PHD protein was incubated with several types of
biotinylated histone H3. Histone H3 was pulled down with
streptavidin beads and GST-PHD was detected with an anti-GST
antibody. Based on the results of the pull-down assay, PHD
interacted weakly with histone H3K4me2 and strongly with
H3K4me3 (Fig. 6a). No interaction was detected when histone
H3K9me1, H3K4me2, H3K9me3, H3K27me1, H3K427me3,
or H3K36me1 was used (Fig. 6a, b). Next, GST-PHD(C162S) or

GST-PHD(C117S) was prepared to confirm whether base
substitution influenced the binding activity of PHD. GST-PHD
(C162S) was not found to interact with histone H3K4me3
(Fig. 6c), suggesting that disrupting binding with histone
H3K4me3 by substitution of cysteine to serine affects the
function of SIZ1, as *SIZ1(C162S):GFP* was unable to complement
the *siz1-2* mutation. Conversely, GST-PHD(C117S) was able to
bind to histone H3K4me3 (Fig. 6c); thus, the *siz1-2* mutation was
complemented by *SIZ1(C117S):GFP*. Because *siz1-2* plants
harboring *SIZ1(C117S):GFP* looked similar to wild-type plants,
the binding activity of PHD(C117S) to histone H3 (Fig. 6c) may
not affect the phenotype of the *siz1-2* mutant harboring *SIZ1
(C117S):GFP*. And difference of binding activity of PHD or PHD
(C117S) to histone H3K4me3 may not be effective for
complementation. For negative control, interaction between
GST and histone H3 or H3K4me3 was examined (Fig. 6d).

**Histone H3K4me3 status in *WRKY70***. WRKY70 is a tran-
scription factor that positively regulates SA-responsive genes, and
its expression is promoted by increased levels of salicylic acid[38].
The *siz1-2* mutant accumulates salicylic acid and exhibits a dwarf-
like phenotype[7,39]. Thus, histone H3K4me3 status in the
*WRKY70* gene was investigated by chromatin immunoprecipita-
tion (ChIP) assay. Under normal conditions, histone H3K4me3
did not accumulate in *WRKY70* in the wild type (Fig. 7a). Under
cold treatment, histone H3K4me3 was detected in the wild type
(Fig. 7a). This is because salicylic acid accumulation was pro-
moted by cold stress, as described previously[40]. On the other
hand, the histone H3K4me3 was highly detected in the *siz1-2*
mutant under both normal and low-temperature conditions
(Fig. 7a). The expression level of *WRKY70* was investigated. The
transcription of *WRKY70* was slightly induced by cold stress
(Fig. 7b). *WRKY70* expression was higher in the *siz1-2* mutant
before and after cold treatment and low expression was observed

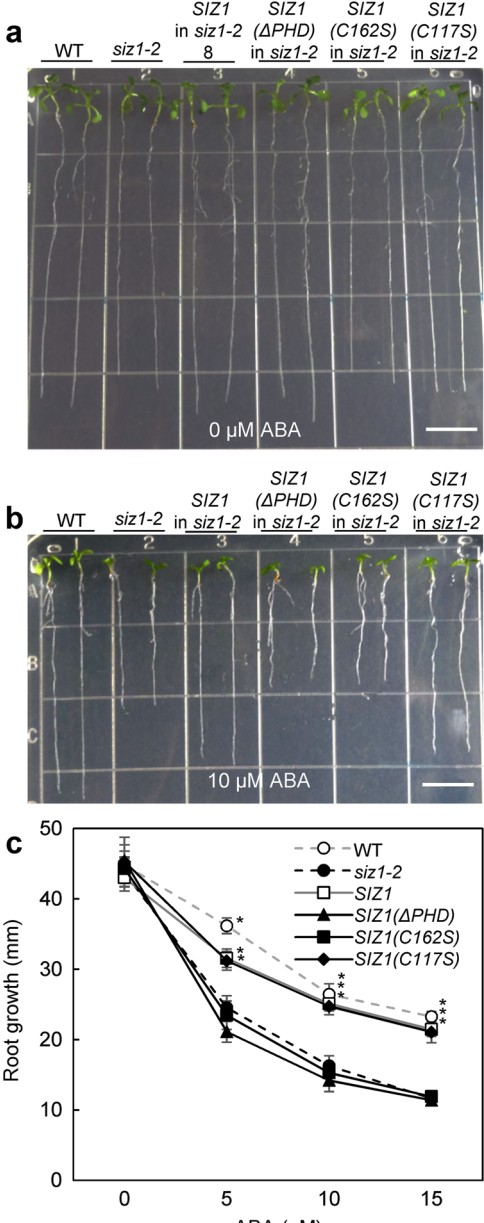

**Fig. 2 Mutation in the PHD finger of SIZ1 was unable to complement ABA inhibition of primary root growth in seedlings of the *siz1* mutant.** Three-day-old seedlings were transferred onto basal media containing 0 (**a**) or 10 μM ABA (**b**). The bar indicates 1 cm length. **c** Root growth values expressed as mean ± standard deviation (SD; *n* ≥ 15 biologically independent seedlings). Asterisk indicates a significant difference from the *siz1* plants (*p* < 0.05) as determined by one-way ANOVA followed by the Tukey–Kramer test.

in the *atx1* mutant (Fig. 7b). The expression level of *WRKY70* was correlated with the status of histone H3K4me3. These results suggest that high levels of histone H3K4me3 in the *siz1-2* mutant enhanced expression of the *WRKY70* gene.

**Interaction between SIZ1 and ATX1**. As shown above, PHD preferentially recognized histone H3K4me3 (Fig. 6), and dysfunction of SIZ1 enhanced H3K4me3 status in the *WRKY70* promoter (Fig. 7). SIZ1 may inhibit methylation of histone H3K4 and prevent transcription initiation via interaction of histone H3K4me3, which is a prominent histone mark associated with

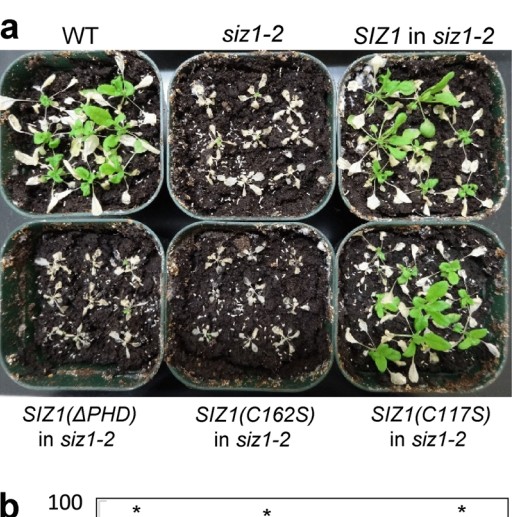

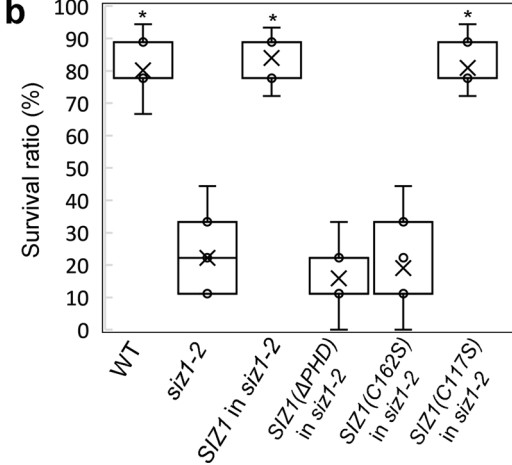

**Fig. 3 Cold sensitivity caused by the *siz1-2* mutation was not recovered by *SIZ1* with mutation in the PHD finger. a** Three-week-old plants were incubated at 4 °C for 1 week for cold acclimation. Cold-acclimated plants were exposed for 4 h at − 6 °C. Following freezing treatment, plants were incubated at 24 °C for 1 week. Photographs are representative of WT, *siz1-2*, and *siz1-2* transformed with *SIZ1* variants. **b** Survival was determined in 27 plants after freezing treatment at − 6 °C. Data are the mean ± SD calculated from four or more independent experiments. Asterisks indicate a statistical difference from the *siz1-2* plants (*p* < 0.05) as determined by Student's *t* test.

transcriptionally active genes[41]. Among histone lysine methyltransferases, ATX1 mediates H3K4 trimethylation and ATX2 mediates H3K4 dimethylation[34,35]. Concurrent disruption of the *ATX3*, *ATX4*, and *ATX5* genes significantly reduced H3K4me2 and H3K4me3 levels, suggesting that these are redundant for H3K4 di- and trimethylation[42]. These previous results suggest that ATX1–5 are involved in the transfer of a methyl group to histone H3K4. Thus, we assumed that the PHD finger of SIZ1 interacts with ATX to block activity of histone lysine methyltransferase. ATX1–5 proteins contain catalytic SET domains, which bind *S*-adenosylmethionine and the substrate lysine[43]. The SET domains of ATX proteins fused with maltose-binding protein (MBP) were purified, and each MBP-ATXSET was incubated with GST-PHD. After pull-down with GST-PHD, MBP-ATXSET was detected with an anti-MBP antibody. All SET domains were detected by western blot analysis (Fig. 8), indicating that the PHD of SIZ1 was able to interact with the SET domain of each ATX protein. Because GST-PHD(C162S) also detected all SET domains, substitution of C162 in the PHD domain did not affect binding activity to the SET domain of each ATX protein.

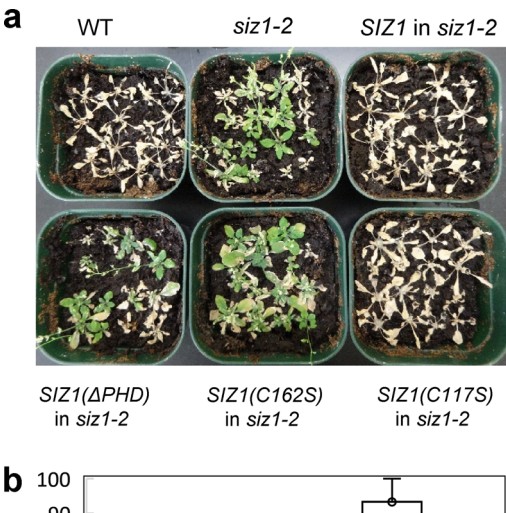

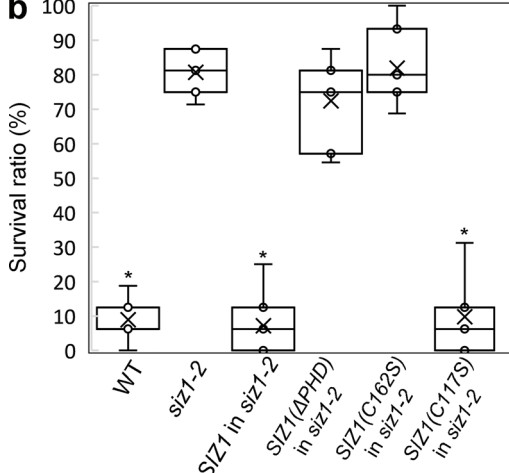

**Fig. 4 Drought-tolerant phenotype of the *siz1-2* mutant is not recovered by *SIZ1* with mutation in the PHD finger. a** Watering was resumed, and plants were incubated for 1 week after the 2-week drought treatment. **b** The survival ratio was determined for 16 plants after drought treatment. Data are mean ± SD from three independent experiments. Asterisks indicate a statistical difference from the *siz1-2* plants ($p < 0.05$) as determined by Student's *t* test. The data are a representative experiment from three independent experiments.

Probably, PHD has different binding activity to histone H3K4me3 or ATX proteins.

To confirm interaction in vivo, full-length coding sequence of SIZ1 including the PHD finger or ATX1 including the SET domain was inserted into pBYR2HS[44] fused with the FLAG tag or the RAP tag, respectively. SIZ1-FLAG or SIZ1(C162S)-FLAG was immunoprecipitated with anti-DYKDDDK antibody. The immunoprecipitant was detected with anti-RAP tag antibody[45]. These results suggest that SIZ1 interacts with ATX1 in vivo.

## Discussion
SIZ1 is an important SUMO E3 ligase in plants, and only plant SIZ proteins contain a PHD finger[46]. In the present study, we demonstrated that this PHD finger is biologically important, because a mutation or deletion of the PHD finger from SIZ1 was unable to complement the *siz1-2* mutation (Figs. 1–4). Furthermore, the PHD finger preferentially recognized trimethylated histone H3K4 (Fig. 6a, b). Substitution of C162 with S prevented interaction between the PHD finger and H3K4me3 (Fig. 6c), suggesting that inhibition of PHD (C162S)-binding activity may prevent complementation of the *siz1-2* mutation. The PHD finger also interacted with ATX proteins (Fig. 8), which methylate

histone H3K4, probably resulting in suppressed methylation function of ATX proteins.

Histone H3K4me3 is a landmark of active gene expression[47]. The recognition of H3K4me3 by PHD was first reported in human BPTF and ING2[26,27,33,48]. PHD recognizes H3K4me3, but it does not affect transcriptional activation or repression, which is dependent on the recruitment of histone acetyl transferase or a histone deacetylase complex. Our results suggest that recognition of H3K4me3 by *Arabidopsis* SIZ1 induced transcriptional repression of *WRKY70*, because the *siz1-2* mutation promoted its expression (Fig. 7). In mammalian cells, sumoylated histone H4, which is associated with gene repression, in nucleosomes activates LSD1 (lysine-specific demethylase1) by a mechanism dependent on the SUMO-interacting motif in CoREST (co-repressor for element 1 silencing transcription factor)[49]. LSD1 demethylates methylated histone H3K4 to downregulate gene expression. Sumoylated histone H4 enhances development of the LSD1–CoREST complex to repress gene activity. It is plausible that SIZ1-bound histone H3K4me3 may recruit demethylase or a co-repressor to repress gene expression in plants.

Although several reports have demonstrated that PHD fingers interact with H3K4me2/3, some bind to other histone marks, such as H3K9me3[50], H3K36me3[51], H3R2[52], or unmodified H3K4[53,54]. PHD fingers are widely conserved in eukaryotic organisms, including plant species. PHD-containing proteins in plants are involved in the regulation of various biological functions, such as pathogen defense responses, developmental processes, and flowering time[55]. These results suggest that PHD has regulatory functions and is involved in the mediation of cross-talk between the epigenetic status of chromatin, and signaling and cellular pathways.

The PHD finger of *Arabidopsis* SIZ1 interacts with the SUMO E2 enzyme AtSCE1 and the bromodomain global transcription factor group E (GTE)[30]. The PHD contributes partially to sumoylation of AtSCE1 but is indispensable for sumoylation of GTE[30]. Similarly, the PHD finger of the human KRAB-associated protein 1 (KAP1) co-repressor binds to the SUMO E2 enzyme Ubc9 and functions as an intramolecular SUMO E3 ligase[56]. The PHD-mediated sumoylation of the adjacent bromodomain is required for gene silencing[56,57]. Conversely, the PHD finger of rice SIZ1 was unable to bind to SUMO E2[29]. These results suggest that the tandem PHD finger-bromodomain, or association between the PHD finger with the bromodomain, is required for transcriptional silencing.

The bromodomain typically recognizes acetylated histones to interpret HAT activity and recognize the histone code[58]. The bromodomain is often found in combination with the PHD finger, as described above. And the bromodomain is also found in combination with other histone-binding domains, including the MBT and WD40 domains[58]. These domains recognize multiple modifications to regulate transcriptional activation or silencing. The NURF chromatin-remodeling complex subunit BPTF harbors a bromodomain and a PHD finger to interact with H4K16ac and H3K4me3, respectively, in the same nucleosome[59]. Conversely, lysine methylation is mediated by SET domain proteins, which are identified in *Drosophila* Su(var)3-9, Ez (Enhancer of Zesta), and Trithorax[60]. Su(var)3-9 methylates histone H3K9, whereas Ez methylates H3K27[61]. Trithorax is more likely to mediate methylation of histone H3K4. Based on similarity, five ATX (*Arabidopsis* trithorax) and two ATX-related proteins (ATXR3 and ATXR7) have been proposed as H3K4 methyltransferases[62]. In fact, ATX1, ATX2, and ATXR7 have been shown to be involved in *FLC* activation and histone methylation[35,63]. To suppress transcription, histone methylation at histone H3K4 should be blocked. The SET domains of ATX proteins interacted with the PHD finger of SIZ1 (Fig. 8). It is

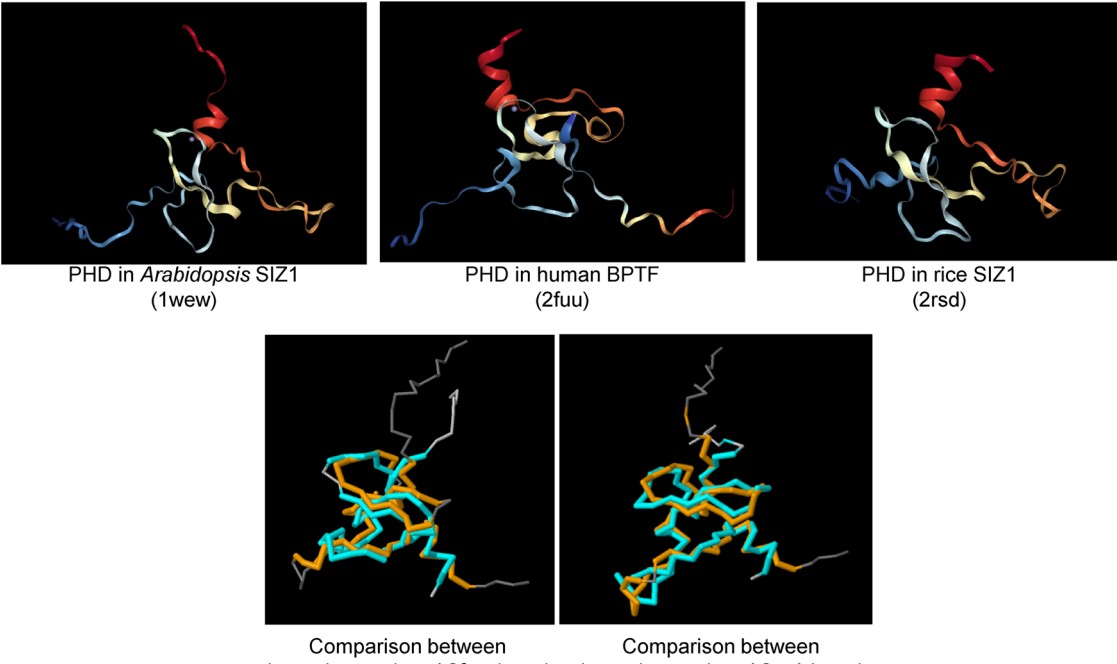

**Fig. 5 Comparison of the 3-D structure of the PHD finger in *Arabidopsis* SIZ1 (PDB ID: 1wew) and human BPTF (2fuu) or the PHD finger in rice SIZ1 (PDB ID: 2rsd).** The comparison was performed in PDB (http://www.rcsb.org/pdb/workbench/workbench.do).

likely that the PHD finger of SIZ1 interacts with ATX proteins to inhibit their access to histone H3K4.

The PHD finger of SIZ1 recognizes histone H3K4me3 and this interaction may suppress transcriptional activation. The PHD finger also binds to ATXs and this interaction may repress methyltransferase activity for further activation of transcription. Because PHD(C162S) was unable to complement the *siz1-2* mutation (Figs. 2–4) and was unable to interact with histone H3K4me3 (Fig. 6) but bound to ATXs (Fig. 8), recognition of histone H3K4me3 is more important for regulation of transcription.

In conclusion, the PHD finger of SIZ1 is important for recognition of the histone code, H3K4me3, to suppress transcription of *WRKY70* and for interaction with ATX proteins, probably to inhibit their binding to histone H3K4.

## Methods

**Plant materials and physiological analysis.** The *Arabidopsis* T-DNA insertion mutants *siz1-2*[15] and *siz1-2* were transformed with pCambia1302-AtSIZ1full:: AtSIZ1:GFP and the resulting plants, *siz1-2* with *Pro_{SIZ1}::SIZ1:GFP*[31] were on the Col-0 background.

To produce *siz1-2* with *Pro_{SIZ1}::SIZ1(ΔPHD):GFP*, the 5′- or 3′-regions of SIZ1 were amplified with the primers, SIZ1-2hyF and SIZ1-PHDdeltaR, or SIZ1-PHDdeltaF and SIZ1-2hyR (Supplementary Table 1), respectively. After purification of PCR products, *SIZ1(ΔPHD)* was amplified with the primers, SIZ1-2hyF and SIZ1-2hyR, with the 5′- and 3′-regions of SIZ1 as templates. The PCR product was digested with *Sal*I and *Pst*I and the resulting product was inserted with *Sal*I- and *Pst*I-digested pCambia1302-AtSIZ1full::AtSIZ1:GFP. The resulting plasmid, pCambia1302-AtSIZ1full::AtSIZ1(ΔPHD):GFP, was transformed into *Agrobacterium* and then transgenic Col-0 plants containing *Pro_{SIZ1}::SIZ1(ΔPHD): GFP* were produced. The transgenic Col-0 plant was crossed with the *siz1-2* mutant. In the F2 population, plants harboring both homozygotes of the *siz1-2* mutation and *Pro_{SIZ1}::SIZ1(ΔPHD):GFP* were selected and named *siz1-2* with *Pro_{SIZ1}::SIZ1(ΔPHD):GFP*. The *siz1-2* mutant or wild-type *SIZ1* was determined with the primers, LP034008 and Salk_Lba1 or LP034008 and RP0034008 (Supplementary Table 1), respectively. Similarly, pCambia1302-AtSIZ1full::AtSIZ1 (C162S):GFP or pCambia1302-AtSIZ1full::AtSIZ1(C117S):GFP was produced with the primers, SIZ1-2hyF, AtSIZ1-C162S-R, AtSIZ1-C162S-F, and SIZ1-2hyR, or SIZ1-2hyF, AtSIZ1-C117S-R, AtSIZ1-C117S-F, and SIZ1-2hyR (Supplementary Table 1), respectively. These vectors were transformed and *siz1-2* with *Pro_{SIZ1}::SIZ1 (C162S):GFP* or *siz1-2* with *Pro_{SIZ1}::SIZ1(C117S):GFP* were produced. *Arabidopsis*

plants were grown in soil or Petri dishes at 24 °C under a long-day photoperiod (16 h light/8 h dark).

To examine the effect of ABA hypersensitivity on root growth, the seeds were surface-sterilized and sown onto half Murashige and Skoog (MS) medium containing 1% sucrose and 0.8% agar. Three-day-old seedlings were transferred onto media supplemented with 0, 5, 10, or 15 μM ABA (Sigma). Root growth was measured as the difference in root length between the beginning and end of the growth evaluation period[24].

To evaluate freezing sensitivity, the plants were grown at 24 °C for 3 weeks in soil and, then incubated at 4 °C for 1 week for acclimation to low temperature. After acclimation, plants were incubated at 0 °C for 1 h, and the temperature was lowered by 2 °C h$^{-1}$ until it reached to $-6$ °C and was maintained for 4 h in the incubator (IN602, Yamato Scientific Co., Ltd., Tokyo, Japan). Then, plants were incubated at 4 °C overnight and transferred to 24 °C. The survival ratio was determined 10 days after the freezing test was performed, as described previously[64].

For the drought-tolerance test, water was withheld from 2-week-old plants for 2 weeks. Then, watering was resumed. To quantitatively determine the survival ratio, 16 plants of each genotype were grown in the same pot. After water was withheld followed by re-watering, the survival ratio was calculated as described previously[4].

These physiological analyses were performed three or more times and representative data were shown in figures.

**RNA preparation and RT-PCR.** Total RNA was isolated using TRIzol reagent (Thermo Fisher Scientific), according to the manufacturer's protocol. RT-PCR was performed as described previously[65]. The primers LP023805 and RP023805 (Supplementary Table 2) were used to detect *SIZ1* expression. The primers as described in the previous work[66] were used for detection of *WRKY70* expression. Data are a representative result from three independent experiments.

**Pull-down analysis.** The PHD region was amplified with the primers pGEX5X-PHD-F and pGEX5X-PHD-R (Supplementary Table 1). The PCR product was inserted into *Eco*RI- and *Sal*I-digested pGEX5X-1 (GE healthcare) via the In-Fusion reaction (Takara Bio). The vector was transformed into *Escherichia coli* Rosetta-Gami(DE3)pLysS (Novagen), and GST-PHD protein was purified with Glutathione Sepharose 4 Fast Flow (GE healthcare) according to the manufacturer's protocol. One-microgram of GST-PHD was incubated with 0.5 μg of biotinylated histone H3 variant (Active Motif) in 300 μL of binding buffer (50 mM Tris-HCl, pH 7.8, 300 mM NaCl, 0.1% NP-40, and 1 mM PMSF) overnight. Biotinylated histone H3 variant was pulled down with streptavidin beads. The sample was separated by sodium dodecyl sulfate polyacrylamide gel electrophoresis (SDS–PAGE), and western blot analysis was performed with anti-GST antibody. For input, 10 ng of GST-PHD was loaded. For loading control, 0.5% of mixture before pull-down was loaded onto SDS–PAGE and GST-PHD was detected by anti-GST antibody.

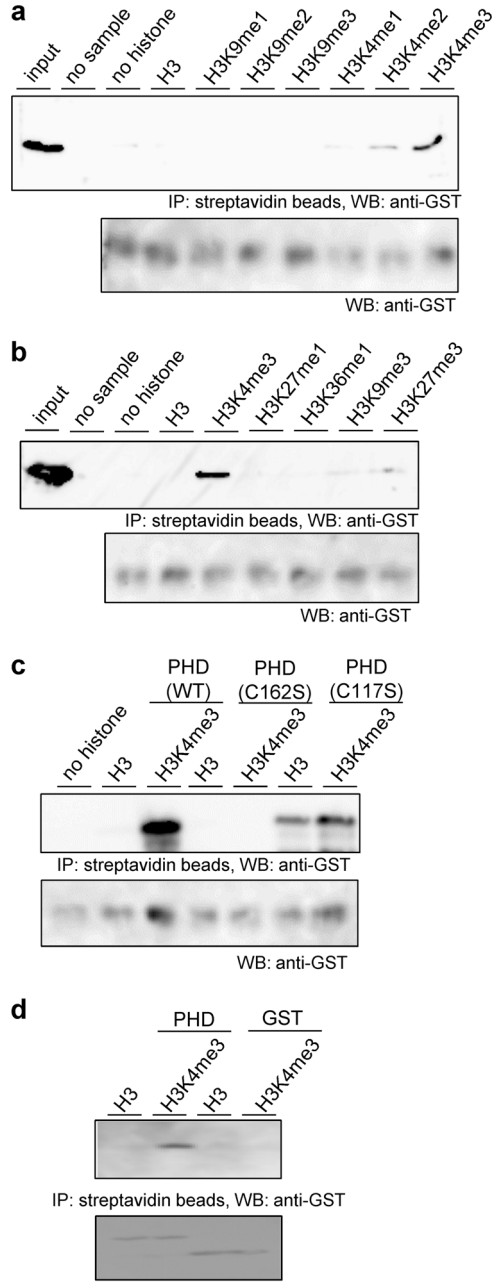

**Fig. 6 The PHD finger of SIZ1 recognizes trimethylated histone H3K4me3.** The PHD finger was inserted into pGEX5X-1 to produce a GST-PHD fusion protein. GST-PHD fusion proteins were incubated with histone H3 or methylated histone H3 fused with biotin. Pull-down of biotin-histone H3 was performed with streptavidin beads. Then, the proteins were detected by immunoblot analysis with anti-GST. **a** Histone H3 or methylated histone H3K4 or H3K9 were used for pull-down experiments. **b** Pull-down analysis was performed with several types of methylated histone H3. **c** The interaction between GST-PHD variants and histone H3K4me3 was investigated. GST-PHD, GST-PHD(C162S), and GST-PHD (C117S) were prepared and pull-down experiments were performed. **d** For negative control, pull-down assay between GST and histone H3 or H3K4me3 was performed. The bottom panel of each figure is a loading control. Unprocessed blots were provided in Supplementary Fig. 4.

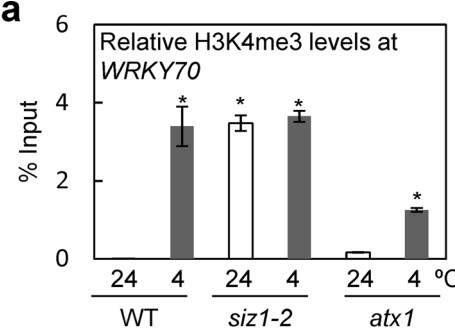

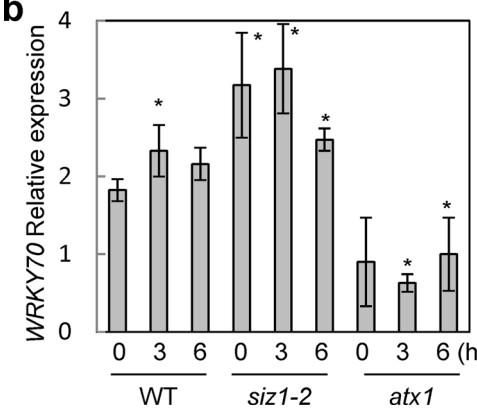

**Fig. 7 H3K4me3 levels and transcription of *WRKY70* in the *siz1-2* mutant.** **a** Levels of histone H3K4me3 at the *WRKY70* gene in wild type and the *siz1-2* mutant. Wild type, the *siz1-2* mutant, and the *atx1* mutant were grown for 3 weeks at 24 °C. The plants were then subjected to cold treatment at 4 °C for 3 h. Then, a ChIP assay was performed. The *atx1* mutant was used as a control to demonstrate that that levels of histone H3K4me3 were low compared with wild type, as described previously[69]. **b** *WRKY70* expression was detected in wild type, the *siz1-2* mutant, and the *atx1* mutant with or without cold treatment. Each experiment was repeated three times and representative data are shown. Each bar represents the standard error of the mean (± SE, $n = 3$). Asterisks indicate a statistical difference from the wild type without cold treatment ($p < 0.05$) as determined by Student's *t* test.

The SET domains of ATX proteins were amplified with the primers, ATX(1 or 2)SET-F-SalI and ATX(1 or 2)-SalI-R (Supplementary Table 1), respectively. The SalI-digested PCR products were inserted into the SalI-digested pMAL-c2X (New England Biolabs) to produce pMAL-ATX1SET and pMAL-ATX2SET. The SET domains of ATX3–5 were amplified with the primers, ATX(3–5)-F-BamHI and ATX(3–5)-BamHI-R (Supplementary Table 1), respectively. The PCR products and pMAL-c2X were digested with BamHI and ligated. The resulting vectors were named pMAL-ATX(3–5)SET. The vectors were transformed into *E. coli* Rosetta-Gami(DE3)pLysS (Novagen), and MBP-ATXSET protein was purified with amylose resin (New England Biolabs) according to the manufacturer's protocol. One-microgram of GST-PHD and 1 μg of MAL-ATXSET were incubated in 300 μL of peptide-binding buffer (50 mM Tris-HCl, pH 7.5, 150 mM NaCl, 0.1% NP-40) overnight. GST-PHD was pulled down with glutathione Sepharose beads. The sample was separated with SDS–PAGE and western blot analysis was performed with an anti-MBP monoclonal antibody (New England Biolabs).

For all immunoblot analysis, the primary antibody was used with 20,000 dilution and the secondary antibody was used with 10,000 dilution.

**Co-immunoprecipitation assay.** The FLAG tag or the RAP tag was introduced into the high-level transient expression 'Tsukuba system'[44,67]. The PCR products containing His-tag (HHHHHH), and FLAG tag (DYKDDDDK) were produced with the primers pBYR2HS-Flag-F, FLAG-His, and pBYR2HS-FlagHis-R (Supplementary Table 1). The PCR products containing His-tag (HHHHHH), RAP tag (DMVNPGLEDRIE)[45], and the recognition site for HRV 3 C protease (LEVLFQGP), were produced with the primers pBYR2HS-HRV3C-F, HRV3C-RAP-His, and pBYR2HS-stopHis-R (Supplementary Table 1). These PCR products were introduced into the SalI-digested pBYR2HS with an In-Fusion HD Cloning

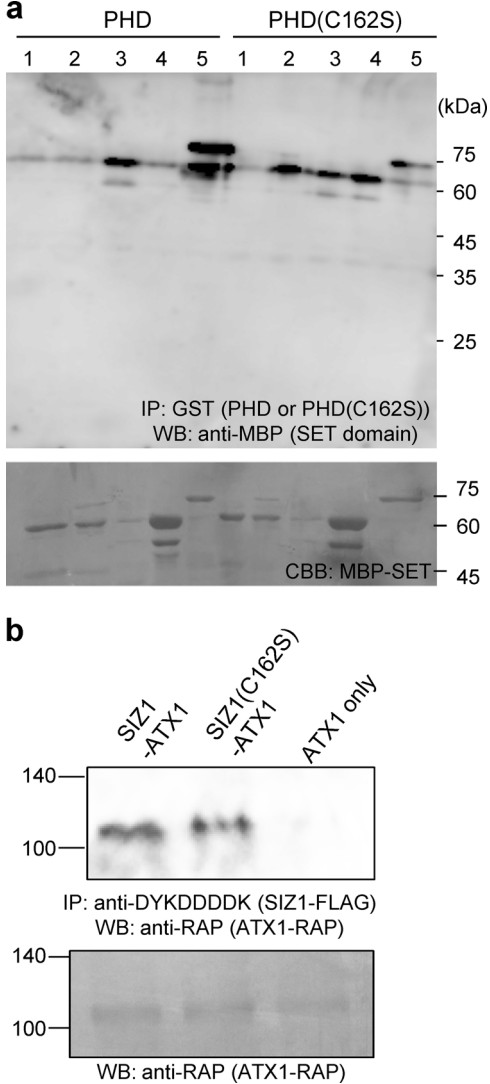

**Fig. 8 The PHD finger of SIZ1 interacts with the SET domain of ATX proteins. a** GST-PHD or GST-PHD(C162S) and MBP-ATXSET were incubated and pull-down of GST-PHD or GST-PHD(C162S) was performed with glutathione Sepharose beads. Then, MBP-ATXSET proteins were detected by immunoblot analysis with an anti-MBP monoclonal antibody. Before immunoprecipitation, 5% of protein mixture was loaded onto SDS–PAGE and the gel was stained with Coomassie Brilliant Blue to detect MBP-SET domain of each ATX protein. **b** Interaction between SIZ1 or SIZ1 (C162S) and ATX1 in vivo. Entire coding sequence of SIZ1 and ATX1, which are fused with FLAG and RAP, respectively, was transiently expressed in *N. benthamiana*. The immunoprecipitation was performed with anti-DYKDDDDK antibody magnetic beads and the immunoprecipitant was detected by anti-RAP antibody. Unprocessed blot was provided in Supplementary Fig. 4.

Kit. The resulting constructs were designated as pBYR2HS-CFH or pBYR2HS-CRH, respectively. The coding sequence of SIZ1 or ATX1 was amplified with the primers, pBYR2HS-AtSIZ1-F and pBYR2HS-AtSIZ1-R, or pBYR2HS-ATX1-F and pBYR2HS-ATX1-R (Supplementary Table 1), respectively. The resulting PCR products were introduced into the *Sal*I-digested pBYR2HS-CFH or pBYR2HS-CRH. pBYR2HS-SIZ1-FH, pBYR2HS-SIZ1(C162S)-FH, or pBYR2HS-ATX1-RH was transformed into *Agrobacterium tumefaciens* GV3101, then, agroinfiltration was performed to *Nicotiana benthamiana*[44]. Soluble protein solution was prepared with lysis buffer[65]. SIZ1-FH or SIZ1(C162S)-FH was immunoprecipitated with anti-DYKDDDDK tag antibody magnetic beads (Fujifilm Wako Pure Chemical Industries). The immunoprecipitant was separated by SDS–PAGE and immuno-blot analysis with an anti-RAP tag antibody (PMab2)[45].

**Chromatin immunoprecipitation (ChIP) assay**. To investigate histone H3K4me3 status in the promoter of *WRKY70*, a ChIP assay was performed as described previously, with modification[68]. Three-week-old wild-type *siz1-2*, and *atx1* plants (SALK_149002C) were treated with or without cold (4 °C) for 3 h, and the seedlings were subsequently harvested and cross-linked with buffer A (0.4 M sucrose; 10 mM Tris-HCl, pH 8.0; 1 mM EDTA; 1 mM PMSF; 1% formaldehyde). The cross-linking reaction was stopped with 125 mM glycine. The tissues were ground in liquid nitrogen, resuspended in lysis buffer (50 mM HEPES, pH 7.5; 150 mM NaCl; 1 mM EDTA; 1% Triton X-100; 0.1% deoxycholate; 0.1% SDS; 1 mM PMSF; and 1 × protease inhibitor cocktail [Nacalai Tesque]), and sonicated (Microson XL-2000 sonicator, Qsonica, LLC., USA) to achieve an average fragment size of 0.1–1.0 kb. After sonication, centrifugation was performed, and the supernatants were incubated overnight at 4 °C with anti-histone H3K4me3 monoclonal antibody (Active Motif). Immunoprecipitation was performed using Magna ChIP A Chromatin Immunoprecipitation Kits (Millipore). Quantitative PCR was performed with THUNDERBIRD SYBR Premix (Toyobo) using a real-time PCR Thermal Cycler Dice (Takara Bio). The primers, WRKY70-ChipF and WRKY70-ChipR (Supplementary Table 2), were used for detection. The relative quantities of immunoprecipitated DNA fragments were calculated as the percentage of input chromatin that was immunoprecipitated using the comparative $C_T$ method. Data are a representative experiment from three independent experiments.

**Statistics and reproducibilty**. Statistical analysis was performed using Excel software (Microsoft) or IBM SPSS Statics. *P* values < 0.05 were considered significant.

**Reporting summary**. Further information on research design is available in the Nature Research Reporting Summary linked to this article.

## Data availability

The authors declare that all data supporting the findings of this study are available within the paper and its supplementary information files.

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

## Acknowledgements

We thank Ms. Kazuko Ito, Ms. Rieko Nozawa, and Ms. Yuri Nemoto at Tsukuba-Plant Innovation Research Center (T-PIRC), University of Tsukuba for technical support. This work was supported by JSPS Grant-in-Aid for Scientific Research on Innovative Areas (JP16H01458 and JP19H04637) and by a Cooperative Research Grant from the Plant Transgenic Design Initiative, Gene Research Center, T-PIRC, University of Tsukuba.

## Author contributions

K.M. designed the study. N.R. and K.M performed most of experiments. T.S. and K.M. contributed reagents and materials and analyzed data. K.M. wrote the manuscript.

## Competing interests

The authors declare no competing interests.
