## [Peer Review File · Communications Biology]

Reviewers' comments:

Reviewer #1 (Remarks to the Author):

In this manuscript, the authors found that PHD domain of AtSIZ1 recognizes H3K4me3, and the C162 is a critical residue for H3K4me3 binding. Internal deletion of PHD or substitution of C162S in PHD impaired proper SIZ1 functions. The authors also found that H3K4me3 and transcription levels of WRKY70 were elevated in *siz1-2* under normal conditions, and PHD domain of SIZ1 interacted with SET domains of ATX proteins *in vitro*. Overall, the research will provide new insight into the SUMO field. However, some of the presented data cannot support their conclusions, or not solid.

Comments:

1. In Figure 4a and 4b, please provide picture of the same set of plants (before and after re-water).
2. In Figure 6, GST should be used as a negative control.
3. In Figure 7b, please analyze WRKY70 expression level under 4 oC in WT and *siz1-2*. Moreover, it would be better if the authors show SIZ1, but not SIZ1[C162S], associates with WRKY70 *in vivo*. I think SUMOylation of histones does not provide any useful information for this paper. The authors may consider delete the Figure 7c.
4. In Figure 8, the authors should provide more evidences to show SIZ1 interacts with ATXs *in vivo*.
5. The authors think PHD domain of SIZ1 recognizes H3K4me3, and recruits ATXs, to prevent histone methylation. To support this hypothesis, the authors should test if SIZ1 mutation enhances ATXs association with WRKY70. Even if this is the truth, I don't understand why SIZ1 prevents ATXs recruitment while PHD domain of SIZ1 already recognized H3K4me3.

Reviewer #2 (Remarks to the Author):

Authors address the mechanistic importance of the PHD subdomain in the all important SIZ1 SUMO E3 ligase in Arabidopsis. SIZ1 has been the backbone of plant SUMO studies, so this is a very important topic. Sumoylation has been associated with histone modification and methylation, but with no significant mechanistic insight, so the current MS brings valid novelty. There is a bit more work on characterization of SIZ1 subdomains, but still the present MS is a good addition to the present state-of-the-art, and definitely advances our knowledge on SIZ1 mode of action. They perform an elegant experiment to resolve how SIZ1(C162S) but not SIZ1(C117S) and the previously studied SIZ1(C134) (Cheong et al 2009) is important for the PHD domain's function. Then they employ what is presently known for human PHD domain literature, and infer on PHD domain importance in Arabidopsis for binding to specific histones and ensuing role in expression suppression. Results from the later part could use additional experimentation (see comments below). Most importantly though, figure 7 and potentially figure 8 still need to resolve some issues.

MINOR

70 Authors describe protein topology of SIZ1 and MM21. Mention also topology of PIAL1/2 and potential function as sumo chain editing (actually leading to their alternative naming as E4 ligases www.plantcell.org/cgi/doi/10.1105/tpc.114.131300)

114 ref 24 also demonstrates importance of PHD for sumoylation

127 don't you mean mutant ProSIZ1::SIZ1:GFP ?

FIG1 436 was unable, just one mutation, NOT were unable

FIG 5 and 6 could be merged to avoid such a high number of figures

FIG 6 For clarity, indicate, in the figure, the antibody used

FIG 7 yy axis legends are too incomplete to enable stand alone interpretation of the graphic. Please also contextualize use of ATX mutant in the main text.

FIG 7 Statistics are missing in figure legend. There is no mention as to the type of statistical comparison and test employed, the statistical results presented are very doubtful, and need to be completely revised.

FIG 7 There is no Fig7c legend nor is it mentioned in the main text. Main text is also difficult to interpret. This reads as a late addition to the MS.

Fig7c Support for sumoylation of His3 based on the blot intensity is weak and I suggest that authors repeat the experiment. There are no proper controls, at least single infiltration with the SUM1 vector is required. Authors may try to use the Abcam AtSUMO1 commercial antibody with ample success in Arabidopsis SUMO research.

Loading controls are missing in all WB experiments.

192 you need to contextualize use of the Q90A mutation in the SUMO peptide

FIG 8 Results would be more solid if a second PPI method was tested. At least the GST-PHD mutant variants which are already available could be tested as well for potential importance for this PPI rather than Histone binding. Perhaps properties are different, which would bring added insight into PHD topology importance.

229 PHD acts

232 suggest not indicate

233-236 rephrase confusing sentence

275 check work of kong et al 2016 (doi: 10.1111/jipb.12509) for SUMO and FLC stability

287 insert insertion mutant references (Salk...) to disambiguate. There is no info regarding the atx mutant.

294 the 5'

315 only 10 um was used in the present study

Reviewer #3 (Remarks to the Author):

The authors showed that the PHD finger of AtSiz1 recognizes tri-methylated histone H3K4 for the function of SIZ1 in abiotic stress responses. This study may be interesting for the researchers in the field, but there are several questions need to be answered.

1. A previous study showed that the PHD finger of the rice Siz1 recognizes tri-methylated histone H3K4. It is necessary to compare the protein structures between the PHD domains of the rice and Arabidopsis SIZ1, for illustrating the similarity and difference of SIZ1 among species.
2. In most of phenotype analysis, one transgenic line of each genotype was used. Two independent lines should be included in all the experiments.
3. The authors showed that the C117S mutation also decreases the affinity with H3K4me3 in figure 6C, as well as this residue resides in the PHD motif, please explain the reason why the C117S mutant completely complements the phenotypes of siz1-2.
4. In figure 7B, the data would be more clear if the expression of WRKY70 in both RT and cold condition in all types of plants including the atx1 mutant.
5. Because the PHD finger of AtSIZ1 may also contribute to its interaction with SCE1, how to

distinguish the effect of mutation on histone recognition and SCE1 interaction.

6. Some typing mistakes need to be corrected throughout the manuscript. For example, "alanine" in line 136 should be "serine".

Dear Reviewers,

We thank all reviewers for their helpful comments and appreciate your efforts. Your comments are very useful to improve our manuscript. We have revised the original manuscript by adding new experimental data. Our messages are written in red.

Reviewers' comments:

Reviewer #1 (Remarks to the Author):

In this manuscript, the authors found that PHD domain of AtSIZ1 recognizes H3K4me3, and the C162 is a critical residue for H3K4me3 binding. Internal deletion of PHD or substitution of C162S in PHD impaired proper SIZ1 functions. The authors also found that H3K4me3 and transcription levels of WRKY70 were elevated in *siz1-2* under normal conditions, and PHD domain of SIZ1 interacted with SET domains of ATX proteins *in vitro*. Overall, the research will provide new insight into the SUMO field. However, some of the presented data cannot support their conclusions, or not solid.

Comments:

1. In Figure 4a and 4b, please provide picture of the same set of plants (before and after re-water).

We did the same experiment with more lines as Reviewer #3 pointed. In Supplemental Figure S3, pictures of the same set of plants (before and after re-water) was provided. To simplify the main text, Figure 4a was deleted.

2. In Figure 6, GST should be used as a negative control.

We performed pull-down assay between GST and histone H3 or H3K4me3 for a negative control. No interaction between them was detected (Figure 6d).

3. In Figure 7b, please analyze WRKY70 expression level under 4 oC in WT and *siz1-2*. Moreover, it would be better if the authors show SIZ1, but not SIZ1[C162S], associates with WRKY70 *in vivo*.

We provided WRKY70 expression in WT and *siz1-2* with or without cold treatment

(Figure 7b).

I think SUMOylation of histones does not provide any useful information for this paper. The authors may consider delete the Figure 7c.

As the reviewer #1 suggested, figure 7c was deleted. We agree that this figure did not provide an important information. We focused on protein-protein interaction between SIZ1 and ATX1 *in vivo* as suggested.

4. In Figure 8, the authors should provide more evidences to show SIZ1 interacts with ATXs *in vivo*.

We performed protein-protein interaction between SIZ1 and ATX1 *in vivo* (Figure 8b). Because we would like to see the protein status close to the nature, we used entire coding sequence of SIZ1 and ATX1, not PHD and SET. Thus, the size of SIZ1 and ATX1 is about 100 kDa and 120 kDa, respectively. We confirm that SIZ1, as well as SIZ1(C162S), interacts with ATX1. This is similar results as shown *in vitro* (Figure 8a).

5. The authors think PHD domain of SIZ1 recognizes H3K4me3, and recruits ATXs, to prevent histone methylation. To support this hypothesis, the authors should test if SIZ1 mutation enhances ATXs association with WRKY70. Even if this is the truth, I don't understand why SIZ1 prevents ATXs recruitment while PHD domain of SIZ1 already recognized H3K4me3.

Our hypothesis is that PHD domain recognizes H3K4me3 this interaction may suppress transcriptional activation. Interaction between SIZ1 and ATX1 may be to prevent ATX1 access to histone H3, not recruit ATXs, and prevent further transcriptional activation. Because SIZ1(C162S) can also interact with ATX1, recognition histone H3K4me3 is much important for repression of transcription. This discussion is provided in the main text (L.278-280).

Reviewer #2 (Remarks to the Author):

Authors address the mechanistic importance of the PHD subdomain in the all important SIZ1 SUMO E3 ligase in Arabidopsis. SIZ1 has been the backbone of plant SUMO

studies, so this is a very important topic. Sumoylation has been associated with histone modification and methylation, but with no significant mechanistic insight, so the current MS brings valid novelty. There is a bit more work on characterization of SIZ1 subdomains, but still the present MS is a good addition to the present state-of-the-art, and definitely advances our knowledge on SIZ1 mode of action. They perform an elegant experiment to resolve how SIZ1(C162S) but not SIZ1(C117S) and the previously studied SIZ1(C134) (Cheong et al 2009) is important for the PHD domain's function. Then they employ what is presently known for human PHD domain literature, and infer on PHD domain importance in Arabidopsis for binding to specific histones and ensuing role in expression suppression. Results from the later part could use additional experimentation (see comments below). Most importantly though, figure 7 and potentially figure 8 still need to resolve some issues.

MINOR

70 Authors describe protein topology of SIZ1 and MM21. Mention also topology of PIAL1/2 and potential function as sumo chain editing (actually leading to their alternative naming as E4 ligases www.plantcell.org/cgi/doi/10.1105/tpc.114.131300)

We provide information of PIAL1/2 functions and cited the paper (L67-68).

114 ref 24 also demonstrates importance of PHD for sumoylation

I agree it, but they did not mention about recognition of H3K4me3 by PHD. This point is an important point for this article.

127 don't you mean mutant ProSIZ1::SIZ1:GFP ?

That is right. *ΔPHD* was deleted in this sentence.

And description about *ProSIZ1::SIZ1:GFP* and *SIZ1pro::SIZ1:GFP* was mixed. Thus, we used only *ProSIZ1::SIZ1:GFP*.

FIG1 436 was unable, just one mutation, NOT were unable

That is modified as suggested.

FIG 5 and 6 could be merged to avoid such a high number of figures

Because we provided a loading control in Figure 6 and also Figure 6d as the reviewer #2 suggested. Thus, it is better to separate between Figure 5 and 6.

FIG 6 For clarity, indicate, in the figure, the antibody used

The information was provided in the figures as suggested.

FIG 7 y axis legends are too incomplete to enable stand alone interpretation of the graphic. Please also contextualize use of ATX mutant in the main text.

FIG 7 Statistics are missing in figure legend. There is no mention as to the type of statistical comparison and test employed, the statistical results presented are very doubtful, and need to be completely revised.

In several papers, such as Ding et al 2012,

<https://journals.plos.org/plosgenetics/article?id=10.1371/journal.pgen.1003111>

they use %input to indicate accumulation of histone H3K4me3. For better understanding, we describe “relative H3K4me3 levels at WRKY70” in the figure 7a. Furthermore, description of statistical analysis was provided in the legend of figure 7 (L497-499).

FIG 7 There is no Fig7c legend nor is it mentioned in the main text. Main text is also difficult to interpret. This reads as a late addition to the MS.

Fig7c Support for sumoylation of His3 based on the blot intensity is weak and I suggest that authors repeat the experiment. There are no proper controls, at least single infiltration with the SUM1 vector is required. Authors may try to use the Abcam AtSUMO1 commercial antibody with ample success in Arabidopsis SUMO research.

192 you need to contextualize use of the Q90A mutation in the SUMO peptide

As the reviewer #1 suggested, figure 7c was deleted. And according to the reviewer #2's comment described below, “At least the GST-PHD mutant variants which are already available could be tested as well for potential importance for this PPI rather than Histone binding”, we focused on interaction between SIZ1 and ATX1 rather than histone binding. For PPI in vivo, it is better to produce whole protein of SIZ1 and ATX1,

because it may reflect protein status close to nature. But they are big proteins (both of them are more than 100 kDa). Thus, we decided to use transient expression.

Loading controls are missing in all WB experiments.

Loading controls were provided.

FIG 8 Results would be more solid if a second PPI method was tested. At least the GST-PHD mutant variants which are already available could be tested as well for potential importance for this PPI rather than Histone binding. Perhaps properties are different, which would bring added insight into PHD topology importance.

For Fig 8, we examined protein-protein interaction between PHD or PHD(C162S) and SET domain of each ATX protein. C162S substitution did not affect binding activity to SET domain of each ATX protein. Probably, different binding activity to histone H3K4me3 and ATXs.

229 PHD acts

232 suggest not indicate

233-236 rephrase confusing sentence

These words and sentence were modified as suggested.

275 check work of kong et al 2016 (doi: 10.1111/jipb.12509) for SUMO and FLC stability

This article is an interesting article to demonstrate that SUMO protease ASP1 regulates flowering time through regulation of FLC stability. Our article demonstrate importance of PHD domain in SIZ1 SUMO E3 ligase and interaction between PHD and ATXs. If we describe Kong's work, it becomes ambiguous for this study. Thus, we did not cite Kong's article.

287 insert insertion mutant references (Salk...) to disambiguate. There is no info regarding the atx mutant.

The information of atx mutant was provided (L392).

294 the 5'

This was modified as suggested.

315 only 10 μ m was used in the present study

As shown in Figure 2C, 5, 10, 15 μ M ABA were treated. Thus, no modification was done in this sentence.

Reviewer #3 (Remarks to the Author):

The authors showed that the PHD finger of AtSiz1 recognizes tri-methylated histone H3K4 for the function of SIZ1 in abiotic stress responses. This study may be interesting for the researchers in the field, but there are several questions need to be answered.

1. A previous study showed that the PHD finger of the rice Siz1 recognizes tri-methylated histone H3K4. It is necessary to compare the protein structures between the PHD domains of the rice and Arabidopsis SIZ1, for illustrating the similarity and difference of SIZ1 among species.

We compared 3D structure of PHD in AtSIZ1 and PHD in rice SIZ1. Those are quite similar (Figure 5).

2. In most of phenotype analysis, one transgenic line of each genotype was used. Two independent lines should be included in all the experiments.

We performed phenotype analyses with two independent lines of each genotype. The data are provided in Supplementary Fig. S1-3.

3. The authors showed that the C117S mutation also decreases the affinity with H3K4me3 in figure 6C, as well as this residue resides in the PHD motif, please explain the reason why the C117S mutant completely complements the phenotypes of siz1-2.

Probably, there is criteria for complementation. If the PHD variants can bind to H3K4me3 to some extent, it can complement the *siz1-2* mutation. Thus, we provided the following sentence, “difference of binding activity of PHD or PHD(C117S) to histone H3K4me3 may not be effective for complementation.”

4. In figure 7B, the data would be more clear if the expression of WRKY70 in both RT and cold condition in all types of plants including the *atx1* mutant.

We provided WRKY70 expression in WT and *siz1-2* as well as *atx1* with or without cold treatment (Figure 7B).

5. Because the PHD finger of AtSIZ1 may also contribute to its interaction with SCE1, how to distinguish the effect of mutation on histone recognition and SCE1 interaction.

If we find that PHD(C162S) interacts with SCE1, we can distinguish between interaction with histone H3K4me3 and SCE1. But we still cannot distinguish between histone recognition and other interacting factors, because we also isolated several PHD finger-interacting proteins. To completely distinguish the effect of mutation on histone recognition and other interacting proteins, we need complete list of PHD finger-interacting proteins and check interaction to all proteins. It is difficult. And investigation of SCE1 interaction may not contribute to this study very much. We suppose that it may for another study.

6. Some typing mistakes need to be corrected throughout the manuscript. For example, “alanine” in line 136 should be “serine”.

This is modified as suggested.

REVIEWERS' COMMENTS:

Reviewer #1 (Remarks to the Author):

My concerns have been adequately addressed in this version and it is now a much better story.

Reviewer #2 (Remarks to the Author):

Authors accommodated most of the serious comments made by reviewers, including validation of mutant phenotypes with 2 independent lines, incorporation of controls, and new analysis for PPI interaction. The current MS definitively improved with the reviewing process, and is now a more cohesive scientific effort.

Fig5 Can you produce a panel with the single rice PHD domain like Arabidopsis and human

Reviewer #3 (Remarks to the Author):

The authors have addressed my concerns, and the mc is suitable for publication.

Dear Reviewers,

We thank all reviewers for their helpful comments and appreciate your efforts. We have revised the original manuscript by producing a panel with the single rice PHD domain, as suggested by Reviewer #2.

REVIEWERS' COMMENTS:

Reviewer #1 (Remarks to the Author):

My concerns have been adequately addressed in this version and it is now a much better story.

Reviewer #2 (Remarks to the Author):

Authors accommodated most of the serious comments made by reviewers, including validation of mutant phenotypes with 2 independent lines, incorporation of controls, and new analysis for PPI interaction. The current MS definitively improved with the reviewing process, and is now a more cohesive scientific effort.

Fig5 Can you produce a panel with the single rice PHD domain like Arabidopsis and human

We added a 3D structure of rice SIZ1 PHD finger in Fig. 5.

Reviewer #3 (Remarks to the Author):

The authors have addressed my concerns, and the ms is suitable for publication.